# MCSNet+: Enhanced Convolutional Neural Network for Detection and Classification of *Tribolium* and *Sitophilus* Sibling Species in Actual Wheat Storage Environments

**DOI:** 10.3390/foods12193653

**Published:** 2023-10-03

**Authors:** Haiying Yang, Yanyu Li, Liyong Xin, Shyh Wei Teng, Shaoning Pang, Huiyi Zhao, Yang Cao, Xiaoguang Zhou

**Affiliations:** 1Department of Automation, Beijing University of Posts and Telecommunications, Beijing 100876, China; yang_haiying@163.com; 2Institute for Grain Storage & Logistics, Academy of National Food and Strategic Reserves Administration, Beijing 100037, China; zhy@ags.ac.cn (H.Z.); cy@ags.ac.cn (Y.C.); 3School of Electronics and Information Engineering, Liaoning University of Technology, Jinzhou 121001, China; 4China-Australia Joint Centre for Postharvest Grain Biosecurity and Quality Research, Beijing 100037, China; 5Sinograin Tianjin Dongli Depot Co., Ltd., Tianjin 300300, China; xinliyong@126.com; 6Institute of Innovation, Science and Sustainability, Federation University Australia, University Drive, Mount Helen, VIC 3350, Australia; shyh.wei.teng@federation.edu.au (S.W.T.); p.pang@federation.edu.au (S.P.)

**Keywords:** *Tribolium* and *Sitophilus*, sibling species, insect detection, geographical strains, Enhanced Convolutional Neural Network, wheat background

## Abstract

Insect pests like *Tribolium* and *Sitophilus* siblings are major threats to grain storage and processing, causing quality and quantity losses that endanger food security. These closely related species, having very similar morphological and biological characteristics, often exhibit variations in biology and pesticide resistance, complicating control efforts. Accurate pest species identification is essential for effective control, but workplace safety in the grain bin associated with grain deterioration, clumping, fumigator hazards, and air quality create challenges. Therefore, there is a pressing need for an online automated detection system. In this work, we enriched the stored-grain pest sibling image dataset, which includes 25,032 annotated *Tribolium* samples of two species and five geographical strains from real warehouse and another 1774 from the lab. As previously demonstrated on the *Sitophilus* family, Convolutional Neural Networks demonstrate distinct advantages over other model architectures in detecting *Tribolium*. Our CNN model, MCSNet+, integrates Soft-NMS for better recall in dense object detection, a Position-Sensitive Prediction Model to handle translation issues, and anchor parameter fine-tuning for improved matching and speed. This approach significantly enhances mean Average Precision (mAP) for *Sitophilus* and *Tribolium*, reaching a minimum of 92.67 ± 1.74% and 94.27 ± 1.02%, respectively. Moreover, MCSNet+ exhibits significant improvements in prediction speed, advancing from 0.055 s/img to 0.133 s/img, and elevates the recognition rates of moving insect sibling species in real wheat storage and visible light, rising from 2.32% to 2.53%. The detection performance of the model on laboratory-captured images surpasses that of real storage facilities, with better results for *Tribolium* compared to *Sitophilus*. Although inter-strain variances are less pronounced, the model achieves acceptable detection results across different *Tribolium* geographical strains, with a minimum recognition rate of 82.64 ± 1.27%. In real-time monitoring videos of grain storage facilities with wheat backgrounds, the enhanced deep learning model based on Convolutional Neural Networks successfully detects and identifies closely related stored-grain pest images. This achievement provides a viable solution for establishing an online pest management system in real storage facilities.

## 1. Introduction

Global food security faces significant challenges, especially with the world’s population expected to surpass 9.1 billion by 2050 [1,2]. The impending population boom, combined with shrinking land and natural resources, complicates efforts to maintain an abundant food supply. Efficiently using agricultural resources and minimizing post-harvest losses—which represent 20% to 40% of total losses, with 55% happening during storage [3]—becomes crucial. Insects contribute significantly to these grain losses, accounting for approximately 10–40% globally [4]. In Asia, around 6% of post-harvest losses stem from inadequate storage, with insects and fungi causing about half of these [5]. These infestations reduce seed germination, alter pH levels, decrease protein content, and lead to other detrimental changes [6]. Grains, especially when stored under conditions conducive to insect reproduction, risk both qualitative and quantitative degradation [7]. High metabolic rates of insects, exacerbated by rising temperatures that can reach up to 45 °C and coupled with increased humidity, encourage microbial and mycotoxin growth [8]. In extreme cases, this heat can even ignite grains, leading to financial losses [7]. Hence, addressing post-harvest grain losses is vital for food accessibility and long-term security. This has propelled interest in insect detection technologies within the grain industry.

Historically, various storage insects have proliferated globally, primarily due to human interventions. Modern global transportation and distribution of crops intensify this problem. *Sitophilus* species, notable pests, have caused widespread damage to crops like rice, maize, barley, and wheat. Damage varies by species: 16.25% in wheat, 8.50% in barley, and 6.25% in maize [9]. These pests, including rice and maize weevils, can severely deplete grain stores. Another significant group, the *Tribolium* species, damages stored products like flours, chocolates, fruits, and grains. This damage reduces product weight, quality, and marketability [10]. *Tribolium* species also release carcinogenic quinones, leading to health complications like allergies and dermatitis [11]. Though they do not typically penetrate intact kernels, damp or damaged ones are vulnerable [11]. When *Sitophilus* species infest wheat, they pave the way for secondary pests like *Tribolium* species [12,13].

*Sitophilus* and *Tribolium* sibling species, sharing similar traits [14,15,16] (see Figure 1), often exhibit differing susceptibility to insecticides, complicating pest control strategies. Importantly, resistance to phosphine is on the rise in *Sitophilus oryzae* and *T. castaneum* [17,18,19,20,21,22,23]. These beetles exhibit distinct habitat preferences and population dynamics [24,25,26]. For instance, adult *T. castaneum* can fly, whereas *T. confusum* is flightless [27]. *S. zeamais* displays increased flight activity [19,20,21,22], with infestations frequently originating in the field, potentially leading to storage losses [23,28]. In contrast, *S. oryzae* tends to climb within grain masses [29]. For effective loss mitigation, precise insect species identification coupled with the application of recommended dosages of insecticides, like phosphine or other related chemicals, is essential to manage resistance.

Early detection of insects is vital for effective management and maintaining the quality of food products. Presently, monitoring largely relies on visual examination of external traits, followed by detailed dissection if anomalies are detected. For example, farms often use manual sampling traps and probes, whereas grain storage and handling facilities opt for manual checks, sieving, and the Berlese funnel method [7]. In recent times, molecular techniques have emerged as highly accurate tools for identifying sibling species, especially in detecting insect body parts or immature stages during port quarantine [30]. However, these techniques involve labor-intensive sample preparation, complex instrument operation, and significant initial investments. Additionally, they are not well-suited for continuous monitoring and inspection in grain storage environments, which are essential for maintaining grain quality on a larger scale. Another important concern is workplace safety, as deteriorated grain, including moldy, aged, insect-infested grain, and debris, can lead to clumping. Strict regulations enforced by the US Occupational Safety and Health Administration (OSHA, 1996) mandate safety measures for bin entry, including full-body harnesses and lifelines to mitigate entrapment risks. Fumigators also face dangers when assessing structural integrity or using fumigants. Grain spoilage generates carbon dioxide and methane, impacting air quality, and other harmful gases may be present even when oxygen levels are sufficient. Our study objectives encompass an evaluation of the robustness of novel models based on ImageNet and laboratory Sitophilus datasets, enhancing their effectiveness in insect pest identification programs. Additionally, we assess the correlations between distinct characteristics influencing sibling species populations, pinpointing geographical nuances to optimize targeted insect pest identification programs. This system is designed to be rapid, cost-effective, reliable, user-friendly, environmentally conscious, and suitable for seamless online inspection, offering tangible benefits to the food industry.

## 2. Materials and methods

### 2.1. Targets 

Three geographic strains of *T. castaneum*, Rfb-CD, Rfb-QH, and Rfb-WH, and two geographic strains of *T. confusum*, Cfb-BJ and Cfb-GD, were used in this study. The field-originating *T. castaneum* geographic strains, Rfb-CD, Rfb-QH, and Rfb-WH, were obtained from grain storage enterprises in Chengdu, Qihe, and Wuhan, China. The *T. confusum* geographic strains, Cfb-BJ and Cfb-GD, were originally obtained from grain storage enterprises in Beijing and Guangdong, China. More than 500 of the geographic strain adults were sieved from the flour before analysis.

### 2.2. Data Collection

#### 2.2.1. Image Acquisition and Dataset

The image data used for the study consisted of two parts. The first part was performed in the laboratory and the second part was conducted in the field at the National Grain Depot in Baoding, China.

Laboratory data acquisition: This part of the image data acquisition was performed using the laboratory image acquisition system established in the previous study [31]. A total of 4878 static images depicting the activities of two types of grain storage pests within food products were recorded. This dataset comprises 2661 images of *T. confusum* and 2217 images of *T. castaneum*. The size of each image was 640 Width × 480 Heigt pixels, with 1–7 pest samples to be identified. 

Field data acquisition: On the morning of 8 July 2021, *T. castaneum* and *T. confusum* were placed in 38 W × 27 L × 3 Hcm trays according to different geographic strains on the surface of 55 W × 24 L × 6 Hm grain bulk. After the pests had dispersed freely within the grain surface, recordings were made using a field monitoring device equipped with 100-watt illumination (DS-2DK8296-BG, Hangzhou Hikvision Digital Technology Co., Ltd., Hangzhou, China). Using a distributed field data acquisition system established at the grain depot [31], a total of 25,222 static images depicting the activities and damages caused by these pests on the grain surface were obtained. This dataset comprised 9775 specimens of *T. confusum* and 15,447 specimens of *T. castaneum*. Each image had dimensions of 1280 pixels in width (720 pixels in height) and included 90–170 pest samples.

To prevent any potential impact on prediction performance due to sample size imbalances in the species categories [32], we excluded images from the dataset that had either too many or too few pest samples in the stored grains. Therefore, 3000 images for the laboratory (1500 images per insect species) and 1000 images for the field (200 images per geographic strain) were used for the target identification image dataset. In addition, automatic cropping to 640 W × 480 H pixels was performed using the Matlab R2018b program, considering the distribution of samples on individual live in images and the computer’s arithmetic power.

#### 2.2.2. Ground Truth Bounding Box

After obtaining the dataset of images depicting pest activity in stored grains, we used the labelImg software to manually annotate and label each pest in the images, generating corresponding *.xml files for individual pest photos. These manually annotated bounding boxes in the training, validation, and test sets precisely defined the true boundaries of objects in the images. As shown in Figure 2, this process involved associating each object with (x,y) coordinates representing the bounding box and class labels. Our primary objective was to leverage these class labels and bounding boxes for the construction of an object detection model, subsequently evaluating its performance on both the validation and test sets. Samples with blurred images caused by the rapid crawling of individual pests during their damaging activities in stored grains and those where over 50% of the pest’s head, pronotum, or elytra was obstructed, were labeled as “DIFFICULT” or left unlabeled and were not included in the model training. In total, 26,806 individual pest activity postures were annotated for training and testing, encompassing various damaging postures pests exhibit in stored grains. In the laboratory segment, there were 894 *T. confusum* and 880 *T. castaneum* individual samples, while in the field, there were 9894 individual samples of *T. confusum* and 15,138 individual samples of *T. castaneum*. This diverse range of pest damaging postures contributes to training a model with a broader understanding of pest behaviors and characteristics.

#### 2.2.3. Data Augmentation and Training Dataset

In order to enhance dataset diversity, bolster the model’s resilience and generalization abilities, and optimize the performance in detecting and identifying small target in-sects, data augmentation techniques were employed during the preprocessing of training samples utilized in the network model [33]. Data augmentation involves applying specific transformations to training images to generate additional training samples from the original images without altering the class labels. These transformations may include translation, rotation, scaling, horizontal flipping, and variations in lighting conditions. In this study, positive and negative 90 degree rotations, as well as vertical and horizontal flips, were employed to enrich further positions and angles in which insects appear in the images. The data can also be improved by combining images from several scenes to form a dataset. Therefore, combining data (laboratory data + field data) were also established in this experiment.

### 2.3. Deep Convolutional Neural Network-Based Approach

Advanced deep neural network (DNN) detectors can be generally classified into two categories: single-stage approaches and two-stage methodologies. In two-stage methods, an initial set of sparsely distributed candidate object boxes is first generated, followed by subsequent processes of classification and regression. These candidate boxes are then consolidated and associated through the utilization of a Region of Interest (RoI) pooling module (Figure 3).

The initial stage of Faster R-CNN commences with the utilization of a foundational network, often a pre-trained Convolutional Neural Network (CNN), to extract a comprehensive set of discriminative features. This CNN proves advantageous for facilitating transfer learning, particularly within the domain of feature extraction. Following this, the resulting feature maps are subsequently input into both the Region Proposal Network (RPN) and the Region of Interest (RoI) pooling module [34].

The RoI pooling module retrieves all N proposed locations, with N set to 600, from the Region Proposal Network (RPN) and subsequently extracts the corresponding Region of Interest (RoI) features from the convolutional feature maps. Following this, it resizes the extracted features to a standardized dimension of 7 × 7 × D, where D represents the depth of the feature maps. This dimension standardization is a preparatory step for the final stage, which involves the Region-based Convolutional Neural Network (R-CNN). Within the R-CNN, precision is further enhanced through the refinement of predicted bounding box coordinates, coupled with the application of a softmax layer to generate the ultimate class label predictions for each respective bounding box location.

The two-stage methods exhibit three advantages over their one-stage counterparts: class imbalance is managed using a two-stage structure with sampling heuristics; object box parameters are regressed using a two-step cascade; and two-stage features are employed to describe the objects.

In this study, we introduce a novel framework built on Faster R-CNN that provides superior accuracy for stored-grain insect detection. Extensive comparative experiments on popular bases, including VGG16, ResNet50, ResNet101 [35], and MCS [31], were conducted to achieve high-efficiency detection accuracy. We further propose several effective techniques to improve the new species dataset’s performance, such as adjusting the anchors’ locations and sizes and using a Position-Sensitive Prediction Model (PSPM) in lieu of fully connected layers.

#### 2.3.1. Transfer Learning

Transfer learning pertains to the capacity of a system to identify and employ the knowledge and competencies acquired from prior domains or tasks in the context of novel domains or tasks. This involves leveraging information from a pre-existing model to enhance a newly developed model [36]. A CNN-based model epitomizes this concept as it can be easily adapted to any region or RGB image, irrespective of varying attributes such as color, lighting, background, or the size and shape of target objects, and does so without necessitating human supervision [37]. One significant application of transfer learning is the pre-trained model. These models offer reusable features that can be incorporated into a new model, eliminating the need to train an entirely new model from scratch. When employing a pre-trained model as a feature extractor, it is possible to acquire the feature map by forwarding the input image through the network up to the designated layer, obviating the necessity of traversing the entire network. We can then extract the values at this stage from an activation or pooling layer and use them as feature vectors.

In this study, transfer learning was applied, utilizing pre-trained models such as VGG16, ResNet50, ResNet101, and MCS, to serve as feature extraction frameworks. The convolutional layers responsible for feature learning in the model architecture were frozen, preserving the learned features during the target recognition of different classes. These features were then applied to the extraction phase of feature maps during the training process for detecting and identifying red flour beetles and confused flour beetles. VGG16 is a commonly used architecture in the VGGNet (Visual Geometry Group Network) network structure. One prominent feature of this approach is that, with the exception of the final pooling layer where the number of channels is doubled, the dimensions of the feature maps are reduced by half following each subsequent pooling layer. Starting from the first group with 64 filters, the filters gradually increase to 512 filters in the final group. The entire structure uses 3 × 3 convolutional kernels, which have the advantage of having fewer parameters while maintaining strong learning capabilities. The architecture comprises five sets of thirteen convolutional layers, five max-pooling layers, and three fully connected layers, with each convolutional layer employing the ReLU activation function. In this study, the fully connected (fc) layers were omitted, retaining only the convolutional layers for the purpose of feature map computation.

ResNet50 and ResNet101 are classification models commonly used in the ResNet (Residual Network) network structure. ResNet50 is predominantly constructed around a microarchitecture referred to as the ‘bottleneck residual module’. This architectural framework incorporates only two pooling layers and employs convolutions with strides exceeding 1 to diminish the spatial dimensions of the output. It encompasses a cumulative count of 50 convolutional layers, succeeded by global average pooling and a fully connected (fc) layer comprising 1000 classes. The bottleneck residual module is composed of three convolutional layers. The initial and concluding convolutional layers employ 1 × 1 convolutions, while the middle convolutional layer employs a 3 × 3 convolution. The quantity of filters learned by the first two convolutional layers is a quarter of the quantity of filters learned by the final convolutional layer. Additionally, a supplementary branch connection, referred to as a shortcut, is established between the input located at the module’s base and the summation operation. Here, we only use the convolutional layers to extract features. We noted how all other residual parameters have remained constant while convolutional layers have risen from 50 to 101. 

MCS is a feature extractor developed based on the VGG16 architecture. It consists of four groups, encompassing a total of 12 convolutional layers, four max-pooling layers, one global average pooling layer, and a feature fusion channel. The ReLU activation function is applied to each convolutional layer. The key feature of this extractor is the inclusion of a feature fusion channel during the feature extraction process. This channel serves to alleviate the diminishment of microstructural feature information induced by the presence of multiple layers of Max-pooling, concurrently bolstering microstructural features through the amalgamation of shallower layers.

In this study, the pre-trained VGG16, ResNet50, and ResNet101 models were trained using the publicly available ImageNet [38] dataset. On the other hand, the MCS structure’s pre-training was conducted using the publicly available ImageNet dataset and our laboratory dataset consisting of rice weevils and maize weevils.

#### 2.3.2. Model Customization

We improved the methods for generating, filtering, and refining candidate bounding boxes in the optimized model architecture (Figure 4). Subsequently, we utilized these improvements to train and test the model on different species of *Tribolium* and *Sitophilus* and geographical strains of the *Tribolium*.

##### Anchor Mechanism and Soft Non-Maximum Suppression (Soft NMS)

Object detection relies significantly on the use of anchor boxes to enhance the speed and efficiency of sliding window-based detection. Anchor boxes represent a predefined set of bounding boxes with dimensions tailored to match those of objects within the dataset. These recommended anchor boxes encompass diverse combinations of object sizes that naturally occur in the dataset, incorporating variations in aspect ratios and scales. The quantity of anchor boxes proposed for utilization at different spatial locations within an image is denoted as ‘k’. In our approach, we employed three distinct scales, with anchor box areas of 8^2^, 16^2^, and 32^2^, coupled with aspect ratios of 1:1, 1:2, and 2:1. As a result, for the VGG16, ResNet, and MCSNet models, each (x, y) coordinate position is associated with k = 9 anchor boxes, as illustrated in Figure 5. In the case of MCSNet Plus, the scales were reconfigured while preserving the same three aspects. Given convolutional feature maps of size W × H (typically around 2400), this results in a total of W × H × k anchor boxes.

The standard procedures involved in training an object detection network encompass several key tasks. These tasks include proposing anchor boxes using established computer vision methodologies or identifying potential anchors within the image, aligning these proposed anchor boxes with plausible ground truth objects, categorizing the remaining anchor boxes as the background, and fine-tuning the input proposals through training. During this training process, the network learns to predict offsets concerning reference boxes, specifically involving Δ_x-center_, Δ_y-center_, Δ_width_, and Δ_height_. These offset values enable the network to optimize the fitting of reference boxes without the need to explicitly predict the exact original (x,y)-coordinates. This circumvents the complexity of encoding strict bounding box ‘rules’ directly into the network, a task often deemed impractical. The creation of reference boxes centers around key points within the feature map generated by the CNN network, which are then mapped back to coordinates in the original image. Figure 5b visually underscores the necessity of anchor boxes. For instance, considering Faster R-CNN as an illustration, the VGG network downscales the input image by a factor of 16 in multiple stages. In this context, a single point on the feature map corresponds to a substantial region of 16 × 16 pixels within the input image (referred to as the receptive field). Restricting object localization solely to this region would inevitably lead to reduced accuracy, and in certain scenarios, objects might not be localized accurately at all. The introduction of anchor boxes addresses this limitation, as each point on the feature map can generate a set of *k* distinct boxes, varying in shape and size (as depicted in Figure 5b). This augmentation significantly elevates the likelihood of encompassing objects, thereby substantially enhancing detection recall. The network subsequently refines these bounding boxes to improve precision.

Prudent management of anchor boxes extending beyond image boundaries is of paramount importance. During the training phase, deliberate exclusion of anchor boxes spanning image boundaries is executed to preclude their influence on loss computations. Neglecting these out-of-bounds instances during training has the potential to introduce a substantial volume of intricate error terms, thereby impeding training convergence. Nonetheless, during the testing phase, we apply a fully convolutional Region Proposal Network (RPN) comprehensively over the image. This operation may generate proposal boxes extending beyond image boundaries, necessitating post-processing to clip these boxes to the image boundary. Some RPN proposals exhibit significant overlap with each other, and to mitigate redundancy, we employ the conventional Non-Maximum Suppression (NMS) method. NMS takes into account specific class scores corresponding to VGG16, ResNet50, ResNet101, and MCSNet. In NMS, the process commences with a list of detection boxes denoted as ‘B’, each associated with scores ‘S’. Subsequently, the highest-scoring detection box, ‘M’, is selected, removed from set ‘B’, and included in the final compilation of detection boxes, ‘D’. NMS further eliminates any boxes within set ‘B‘ that exhibit an overlap with ‘M‘ exceeding the predefined threshold value ‘$N_{t}$’. This iterative procedure is reiterated for the remaining boxes in ‘B,’ with the default threshold value for NMS, ‘$N_{t}$’, set to 0.3. In the male-produced aggregation, pheromones of *Tribolium* spp., and *Sitophilus* spp., it appears that males and females move around in the grain mass at low population levels. While feeding, the males release a pheromone that attracts other males and females [39]. For example, aggregations of three or more weevils are commonly observed at a single-grain kernel. Nonetheless, as per the algorithm’s design, situations may arise where objects falling within a predefined overlap threshold could lead to instances of missed detections. In response, we introduce the Soft-NMS algorithm for enhancing MCSNet Plus. This innovative algorithm progressively diminishes the detection scores of all other objects, based on their degree of overlap with ‘M’, representing the highest-scoring detection box. The gradual reduction in scores for overlapping detection boxes with ‘M’ presents a promising approach for refining the traditional NMS method. It is equally evident that detection boxes exhibiting greater overlap with ‘M’ should undergo more pronounced score reduction due to their increased susceptibility to false positives. Consequently, we put forth the subsequent guidelines for refining the pruning procedure.
(1)$$s_{i}=\left\{\begin{array}{c} s_{i},\;iou\left(M,b_{i}\right)<N_{t}\\ s_{i}(1-iou\left(M,b_{i}\right)),iou\left(M,b_{i}\right)\geq N_{t}\; \end{array}\right.$$

Here, $B=\left\{b_{1},\ldots ,b_{n}\right\}$ and $S=\left\{s_{1},\ldots ,s_{N}\right\}$. B denotes the initial array of detection boxes, while S encompasses their associated detection scores. The term $N_{t}$ represents the NMS (Non-Maximum Suppression) threshold, which we establish at 0.3 when employing a linear weighting function. Consequently, detection boxes positioned farther from the highest-scoring detection box, denoted as ‘M’, remain unaffected, while those in close proximity to ‘M’ incur a more pronounced penalty.

Regarding overlap, it exhibits a discontinuous behavior, leading to an abrupt penalty once the NMS (Non-Maximum Suppression) threshold $N_{t}$ is reached. Ideally, a continuous penalty function would be more desirable to avoid sudden fluctuations in the ranked list of detection outcomes. A continuous penalty function should refrain from penalizing non-overlapping cases while applying substantial penalties to high-overlap scenarios. Furthermore, as the degree of overlap decreases, the penalty should gradually intensify, ensuring that the influence of ‘M’ on boxes with minimal overlap remains minimal. Nevertheless, when the overlap between ‘$b_{i}$’ and ‘M’ approaches unity, ‘$b_{i}$’ should incur a conspicuous penalty. With these considerations in mind, we propose updating the pruning step using a Gaussian penalty function in the following manner.
(2)$$s_{i}=s_{i}e^{\frac{{iou\left(M,b_{i}\right)}^{2}}{\sigma }},\forall b_{i}\notin D$$

Here, σ represents the attenuation coefficient, while D denotes the final set of detections. We have opted for a Gaussian weighting function with σ set at 0.5. It is worth noting that the application of NMS does not extend to every detection box; rather, it selectively prunes those that fall below a minimum threshold during each iteration. Consequently, the computational overhead associated with this step remains modest, and it has no discernible impact on the operational runtime of the current detector.

##### Position-Sensitive Prediction Model (PSPM)

Individual sample sizes are extremely small in detecting and identifying stored-grain pests, typically ranging from 8^2^ to 42^2^ pixels. Consequently, when conducting feature extraction through multiple convolutional layers, successive downsampling can result in the gradual loss of positional information for insects within the image [40]. Additionally, due to the small size of these pests, even a few pixels’ offset in the predicted bounding boxes can result in significant errors in the detection and identification results. Several studies have elucidated a significant class of deep neural networks tailored for object detection [34,41,42]. These networks are distinguished by their division into two distinct sub-networks facilitated by a Region of Interest (RoI) pooling layer [42]. The division entails the following: (i) A shared ‘fully convolutional’ sub-network, functioning independently of the RoIs; (ii) An RoI-specific sub-network that conducts computations exclusive to the region of interest. This segregation can be traced back to early classification architectures, typically composed of two sub-networks—one being a convolutional sub-network culminating in spatial pooling layers and the other comprising several fully connected (fc) layers. Consequently, the natural alignment of the final spatial pooling layer in image classification networks with the RoI pooling layer in object detection networks has been established [34,41,42,43].

In this study, we developed a Fully Convolutional Network (FCN) for object detection. The feature maps were shared by detection and classification parts [43]. To account for translational variances in the Fully Convolutional Network (FCN), we created a set of position-sensitive score maps as the FCN’s output. Each score map encodes information pertaining to relative spatial relationships, such as ‘left of the object’. Beyond the FCN, we introduced a position-sensitive Region of Interest (RoI) pooling layer tasked with consolidating data from these score maps, devoid of any subsequent weighted (convolution/fc) layers. The entire architectural framework is trained end-to-end. All trainable layers are convolutional and shared across the entire image, effectively capturing the spatial information essential for object detection. The last convolutional block comprises 256 dimensions, and we initialize a 512-dimensional 1 × 1 convolutional layer for dimension expansion. Subsequently, we apply a $k^{2}(C+1)$ channel convolutional layer, where C denotes the object category count (+1 for background), yielding score maps. These maps are then employed to process the stored small-sized cereal insects, utilizing a k×k=3×3 kernel. To explicitly encode positional cues within each Region of Interest (RoI), we partition each RoI rectangle into k×k grids. Given an RoI rectangle of dimensions w×h, each grid bin’s size approximates to ≈ $\frac{w}{k}\times \frac{h}{k}$ [41,42]. In our methodology, the final convolutional layer generates $k^{2}$ score maps for each object category. Within the (*i*, *j*)-th bin (0≤i; j≤k−1), we define a position-sensitive RoI pooling operation, selectively extracting information from the (*i*, *j*)-th score map.
(3)$$r_{c}\left(i,j|\Theta \right)=\sum_{\left(x,y)\in bin(i,j\right)}z_{i,j,c}\left(x+x_{0},y+y_{0}|\Theta \right)/n$$

Here, the variable $r_{c}(i,j)$ represents the collection response of the c-th category within the (*i*, *j*) partition. In contrast, $z_{i,j,c}$ designates one of the $k^{2}(C+1)$ score maps within the set. The coordinates $\left(x_{0};y_{0}\right)$ correspond to the top-left corner of the Region of Interest (RoI) and n denotes the count of pixels within this partition. The symbol Θ encompasses all trainable parameters of the network. The range of the (*i*, *j*)-th bin is explicitly defined as $\left\lceil i\frac{w}{k}\right\rceil \leq x<\left\lceil (i+1)\frac{w}{k}\right\rceil$ and $\left\lceil j\frac{h}{k}\right\rceil \leq y<\left\lceil (j+1)\frac{h}{k}\right\rceil$. Figure 6 visually elucidates the computational procedure outlined in Eqn. (3), where distinct colors signify various (*i*, *j*) pairs. Eqn. (3) primarily employs average pooling, the prevailing method within this study, though it is also adaptable for performing max-pooling.

In this particular scenario, we have a collection of $k^{2}$ position-sensitive scores, which are then combined for the purpose of Region of Interest (RoI) voting. In the context of this paper, this voting operation is conducted by taking the average of these scores, resulting in the creation of a (C+1)-dimensional vector for each RoI: $r_{c}(\Theta )=\sum_{i,j}r_{c}\left(i,j|\Theta \right)$. Following this, we compute a softmax response across different categories as follows: $s_{c}(\Theta )=e^{r_{c}\left(\Theta \right)}/\sum_{c^{\prime }=0}^{c}e^{r_{c^{\prime }}\left(\Theta \right)}$. These values serve the purpose of evaluating cross-category entropy loss during the training phase and ranking RoIs during the inference process.

In addition to the previously mentioned $k^{2}(C+1)$-dimensional convolutional layer, we have incorporated an additional convolutional layer with dimensions of $4k^{2}$ for the purpose of bounding box regression. Our approach involves position-sensitive RoI (Region of Interest) pooling on this $4k^{2}$-dimensional feature map, resulting in a $4k^{2}$-dimensional vector being generated for each RoI. Subsequently, these vectors are aggregated into a unified four-dimensional vector using an averaging method. Adhering to the parameterization strategy employed in Fast R-CNN, this four-dimensional vector is utilized to parameterize the bounding boxes, represented as $t=(t_{x},t_{y},t_{w},t_{h})$. To maintain simplicity, our bounding box regression is class-agnostic. Nevertheless, it is important to mention that class-specific regression techniques (i.e., involving a $4k^{2}$-dimensional output layer) can also be applied.

Illustrated in Figure 6, the computation involves a shared feature map between the Region Proposal Network (RPN) and the Position-Sensitive Prediction Module (PSPM). Following this, the RPN module suggests Regions of Interest (RoIs), while the PSPM module evaluates category scores and conducts concurrent bounding box regression using these RoIs as a foundation. Notably, there are no trainable layers following the RoI stage, resulting in nearly negligible computational overhead for region computation and leading to accelerated speeds in both training and inference phases.

### 2.4. Metric and Evaluation

The main evaluation indicators used are Intersection over Union (IoU), Average Precision (AP), mean Average Precision (mAP), Recall, Confusion Matrix, and the standard division (std) of the ten-fold cross-validation experiments [44,45]. 

Intersection over Union (IoU), also referred to as the Jaccard Index, stands out as the predominant evaluation metric widely employed in tasks involving segmentation, object detection, and tracking. Object detection tasks are typically composed of two primary objectives: localization, which involves pinpointing the precise position of an object within an image, and classification, which entails assigning a specific category to the detected object. Notably, IoU exhibits an attractive property known as scale invariance. In essence, this property implies that IoU takes into consideration the width, height, and spatial placement of two bounding boxes. Moreover, Normalized IoU focuses on the geometric area of the objects, rendering it insensitive to variations in their sizes. The IoU is a normalized metric that takes values between [0,1] and has good metric properties. It is 0 when two objects have no intersection and 1 when they completely overlap. Hence, IoU proves to be highly suitable for evaluating the conformity between predicted frames and ground truth frames, enabling precision regulation through confidence thresholds. Utilizing IoU aids in ascertaining the efficacy of a detection (true positive). Computing Intersection over Union can, therefore, be determined via
(4)$$IoU\;(\;p,g)=\frac{area\;of\;overlap\;}{area\;of\;union}$$

Here, ‘p’ corresponds to the predicted bounding box, ‘g’ refers to the ground truth bounding box, and Figure 7 provides a visual representation of the IoU computation process.

Following the computation of IoU, it is divided by the total number of class labels associated with the specific category to derive the Average Precision (AP). Average Precision (AP) quantifies the area beneath the Precision–Recall curve, with values falling within the range of 0 to 1. AP corresponds to the mean precision calculated across all recall values spanning from 0 to 1.

Mean Average Precision (mAP) value is computed as the average of the Average Precision (AP) scores for each class across the entire test dataset. It quantifies the percentage of correctly identified positive predictions relative to all relevant ground truths, as expressed in Equation (5). Precision assesses the model’s capability to exclusively recognize pertinent objects, representing the proportion of accurate positive predictions, as defined in Equation (6).
(5)$$Recall=\frac{TP}{TP+FN}=\frac{TP}{all\;ground\;truths}$$
(6)$$Precision=\frac{TP}{TP+FP}=\frac{TP}{all\;detections}$$

Here, TP (True Positives) signifies accurate detections, indicating an IoU value ≥ specified threshold. FP (False Positives) are instances of incorrect detection, characterized by IoU values < threshold. FN (False Negatives) denote ground truth objects that remain undetected. TN (True Negatives) are not utilized as an evaluation metric and pertain to corrected false detections. The choice of threshold is contingent upon the evaluation criterion and is typically set at levels such as 50%, 75%, and 95%.

Confusion matrix serves as a pivotal tool for evaluating the effectiveness of a classifier’s predictions, particularly when the classifier’s primary objective involves assigning class labels to individual input instances. This matrix stands as a classical and widely employed metric for assessing accuracy, holding a prominent position among decision metrics within the realm of supervised machine learning. In the confusion matrix, the columns correspond to the predicted class outcomes, while the rows depict the actual class outcomes. It finds common usage as a fundamental measure for evaluating the performance of classification models. Within this matrix, one can discern the quantities of true positives (TP), true negatives (TN), false positives, and false negatives (FN) generated by the model during its evaluation on test data [46]. Specifically, the diagonal elements reflect instances where the predicted label aligns with the true label, whereas any deviations from this diagonal signify misclassifications made by the classifier. A greater concentration of high values along the diagonal of the confusion matrix signifies a heightened level of prediction accuracy.

### 2.5. Hyperparameters and Environment 

To ensure the model’s capacity for generalization, mitigate the risk of overfitting, and optimize its predictive performance, this experiment adopts a ten-fold cross-validation approach for both training and testing the dataset. The dataset is divided into ten equitably sized subsets, with nine of these subsets employed for model training and validation, while the remaining subset is exclusively designated for model evaluation. This procedure is iterated 10 times, and the model’s assessment is based on the average of the 10 test results and their corresponding standard deviation.

During the training of the Region Proposal Network (RPN), we assign binary class labels (object or non-object) to each anchor. Positive labels are assigned to two categories of anchors: (i) Those with the highest Intersection over Union (IoU) overlap with any ground truth box; (ii) Those with an IoU overlap greater than 0.7 with any ground truth box. It is noteworthy that a single ground truth box may assign positive labels to multiple anchors. Typically, the second condition alone suffices to identify positive samples. If the IoU ratio of a non-positive anchor with all ground truth boxes remains below 0.3, we designate it as a negative anchor. Anchors that do not fall into the positive or negative category do not contribute to the training objective. Following this, we employ Soft Non-Maximum Suppression (NMS) to eliminate redundant proposal Regions of Interest (RoIs), and the final RoIs are produced for subsequent testing.

In the classification process, we define positive RoIs as those with an Intersection over Union (IoU) overlap of at least 0.5 with the ground truth box; otherwise, they are classified as negatives. RPN and Faster R-CNN undergo independent training and modify their convolutional layers differently. Consequently, we adopt an alternating training approach, wherein both networks share convolutional layers instead of being trained as separate entities. This iterative process continues for multiple cycles. We employ a learning rate of 0.0005 for 100 epochs with a learning rate decay of 0.1. Additionally, we utilize a momentum of 0.9 and a weight decay of 0.0005. To update parameters [47], we employ the Adam optimization algorithm with adaptive momentum estimation. Our implementation is based on PyTorch.

The applications were written using Python3.7. CUDA11.2 and Pytorch 1.2.0 deep learning libraries were used. Model training was performed using an 64 GB (DDR4) workstation (Y920, Lenovo, Beijing, China) with an 8 GB(GDDR5) GPU (GeForce GTX 1070, NVIDIA, Santa Clara, CA, USA). 

## 3. Results

Firstly, a comparison and selection of models were conducted based on a laboratory dataset of the *Tribolium* and a mixed dataset combining laboratory and real storage images. Secondly, taking into account the area occupied by storage pests in the captured grain surface images, the size and quantity of anchor boxes were redesigned. Subsequently, the enhanced MCSNet+ model detected three datasets for the *Tribolium* and *Sitophilus* sibling species. Finally, the performance of MCSNet+ was further validated on different geographical strains of the *Tribolium*. The inter-species recognition rate (mAP) of MCSNet+ reached over 92.67 ± 1.74%, with a detection speed below 0.240 ± 0.004 s/img. The performance on geographical strains was slightly lower but achieved above 84.71 ± 1.33%. MCSNet+ successfully meets the practical requirements for stored-grain pest detection and identification.

### 3.1. Comparison of Detection Results of Different Backbone Networks and Datasets

Despite the availability of fully convolutional detectors, investigations revealed that they did not attain acceptable accuracy. We evaluated the following fully convolutional strategies, VGG16, ResNet50, ResNet101, and MCS. When using ImageNet as model pre-training, these network architectures performed on lab + field test have mAPs of 91.83 ± 0.84%, 91.28 ± 0.83%, 90.47 ± 0.99%, and 92.14 ± 0.67%, respectively, higher than that on the field test. We conducted a comparative analysis between our approach and MCS, a formidable competitor that demonstrated superior performance within our laboratory’s insect dataset, specifically among the two *Sitophilus* sibling species (lab SOSZ W.) (Table 1). In contrast to models pretrained on ImageNet, we observe a marginal reduction in average recognition speed; however, there is a concurrent improvement in mAPs (mean Average Precisions) ranging from 1.76–3.71%. These findings indicate that models pretrained on datasets with similarities to the target dataset and subsequently employed for detection tasks on similar data achieve enhanced performance.

We also observed that the mAPs of ResNet50 were approximately 1% higher than those of ResNet101 on both datasets (Table 1). These findings showed that more convolutional layers do not necessarily guarantee better recognition of small objects. As the convolutional layers increase, downsampling gradually causes the loss of discriminative fine-grained details in small objects [40]. This was a consideration in the design of MCS. Furthermore, the denser model ResNet101 exhibited a slower overall prediction speed than the less-dense model VGG16, with a difference of nearly 0.08 s/img. Lastly, the number of targets in each image affects the prediction speed. In the real storage data, each grain image typically contains approximately 20–50 pests, resulting in the fastest prediction speed of 0.319 ± 0.015 s/img. On the other hand, in the laboratory data, each grain image contains only 1–4 pests, leading to a prediction speed of 0.268 ± 0.006 s/img for the combined laboratory and real storage data.

### 3.2. Experimental Results of Enhanced Model

The network is recommended to produce classification scores and regression offsets for anchor boxes of varying scales and aspect ratios at every pixel within the convolutional feature map. The number of anchor scales can affect model classification. To confirm this hypothesis, we varied the anchor scale to the network with a fixed aspect ratio (1:1,1:2, and 2:1). Table 2 shows that smaller input window size from more anchor scales leads to better performance. Also, the performance is seen to saturate beyond a certain pixel, e.g., 48^2^. It also can be seen that mAP appears marginally better with a small pixel. This may also be expected because storage pests occupy a small area in grain images, and by using 4^2^ pixels, insect images are not scaled, allowing for the preservation of features on the insect’s body at a higher resolution. However, the benefit effectively disappeared when the anchor scales were 4^2^, 8^2^, 16^2^, 24^2^, 32^2^, and 48^2^ as the RoI generation speed and predicted speed significantly increased by 1.5 and 1.3 times, respectively. In this context, to effectively detect small-sized storage pests in grain images within an improved model, we specifically computed scales (8^2^, 16^2^, 24^2^, 32^2^, and 48^2^) and aspect ratios (1:1, 1:2, and 2:1) through computer processing. This was done to ensure that the input images have a sufficiently effective receptive field to detect small insects in the grain. These anchor configurations serve as initial approximations in the subsequent two interconnected regression layers, both of which are fully connected.

As illustrated in Figure 8b, an increased quantity of anchor boxes displays overlaps with the ground truth bounding boxes, particularly when the targets are situated nearby. This method of readjusting the anchor boxes based on the size characteristics of the detection targets helps alleviate missed detections caused by the proximity or limited connectivity of stored-grain pests. It enhances the performance and target recall rate of the detection model, reaching a maximum of over 98%. Mean Average Precision (mAP) provides an intuitive evaluation of a network model’s overall performance in terms of object localization and classification. However, it does not explicitly indicate where the performance bottleneck lies and may not accurately describe the training process. On the other hand, the loss function quantifies the disparity between the model’s predictions and the ground truth values, indicating the distance between the predicted values and the actual ones. A lower loss value indicates a smaller deviation between the predicted and actual values, indicating better model convergence. Particularly within the domain of localization regression tasks, the loss function assumes a pivotal role in evaluating the model’s performance.

The loss function encompasses both classification and regression components The classification loss reflects the system’s performance in predicting object classes, while the regression loss guides the network in generating accurately positioned regression boxes. Figure 9 illustrates the loss function curves of MCSNet and MCSNet+ on the training set for the Region Proposal Network (RPN) and Region of Interest (RoI). Overall, the curves of both models exhibit desirable variations, converging rapidly in the initial stages of training and gradually reaching a balanced state as training progresses. Upon comparison, it is noted that the loss functions for both the RPN and RoI of MCSNet range between 0.1 and 0.23, with the classification loss of the RPN showing more significant fluctuations. By improving the model, we have enhanced object localization accuracy, resulting in a significant increase in detection recall. MCSNet+ demonstrates faster convergence, smaller deviations between predicted and actual values, better data fitting, and, thus, improved accuracy in object localization and recognition.

The mAP values for the *Tribolium* testing outperform those for the *Sitophilus* testing by 0.35–2.47%, reaching a maximum of 96.83 ± 0.43% (Table 3). Additionally, the *Tribolium* demonstrates faster prediction speeds compared to the *Sitophilus*. Furthermore, when comparing the three datasets, it can be observed that the mAP values for all four species based on the laboratory dataset are higher than those based on the real storage dataset, with faster testing speeds. This suggests that image resolution impacts the results of object detection. However, it is worth noting that the differences among the three datasets of *Sitophilus* are not significant, which may indicate that increasing the sample size can enhance the precision of detection to some extent.

### 3.3. Tribolium Geographical Strain Identification

The detection and recognition rates of MCSNet+ for different *Tribolium* geographical strains are lower than the inter-species results (Table 4). The mAP values for the two geographical strains of the confused flour beetle decreased by 10.51%, and the mAP values for the three geographical strains of the red flour beetle decreased by 7.03%. This difference can be attributed to the larger inter-species variations than the differences among geographical strains. Additionally, the dataset size for individual geographical strains is smaller than for different species.

In the realm of predictive analysis, a confusion matrix, often referred to as a confusion table, is a tabular representation with a 2 × 2 structure. It tabulates counts for true positives, false negatives, false positives, and true negatives. This matrix facilitates a more comprehensive analysis that extends beyond mere assessment of correct classification proportions or accuracy. When dealing with an unbalanced dataset—where the numbers of observations across different classes vary greatly—relying on accuracy can lead to misleading results. Figure 10 depicts the percentage of correct classifications for various *Tribolium* species and geographical strains, presented as a confusion matrix. The species detection performance is robust, with an accuracy rate of 96.5% (Figure 10a). However, the performance decreases when identifying geographical strains of *T. confusum* and *T. castaneum*, with respective accuracy rates of 92.1% and 90.1% (Figure 10b,c). For *T. castaneum*, many Rfb-QH samples are misclassified as Rfb-WH. Nearly 8% of Rfb-CD samples are misclassified as either Rfb-QH or Rfb-WH. 

This suggests that certain differences exist among different geographical strains of the pest populations. The isolation in different geographic regions exposes them to varying selection pressures, such as local pesticide use, temperature, and humidity conditions, leading to distinct biological characteristics and genetic variations [48]. However, with increased grain trade and transportation between different locations, geographical isolation becomes less apparent and pests can spread through transportation, thereby reducing the differences among geographical strains. In cases where it is necessary to distinguish between geographical strains, this method can serve as an auxiliary tool in the identification process.

## 4. Discussion

The integration of machine vision technology for monitoring stored grains plays a pivotal role in addressing post-harvest losses, ensuring secure storage, and advancing sustainable agricultural practices. This technology enables the tracking of trends in changes in insect populations and infestation levels over defined timeframes. Moreover, it offers valuable insights into insect behavior within diverse environmental conditions and serves as an effective means to evaluate the efficacy of pest control measures. Machine vision represents an emerging technology that amalgamates mechanical, optical instrumentation, electromagnetic sensing, digital, and image processing techniques [49]. It operates by extracting information from images captured by cameras, enabling subsequent object recognition and classification [50]. This technology offers a rapid, consistent, cost-effective, and objective detection method with the potential for assessing the quality of agricultural products. The non-destructive nature of machine vision, along with its speed and precision, aligns seamlessly with the increasing demands for production and quality, thus contributing to the advancement of automated processes. Machine vision technology encompasses three primary processes: image acquisition, image processing or analysis, and identification and interpretation. On occasion, preprocessing is employed to enhance image quality by suppressing undesirable distortions or noise while enhancing crucial visual features of interest [51]. Image feature extraction involves the derivation of image features of varying complexity from image data [52].

### 4.1. Technology 

In this study, we observed a superior recognition rate and speed for *Tribolium* across various models and datasets compared to *Sitophilus*. This discrepancy could likely be attributed to the differences in feature extraction, a crucial aspect impacting the final identification results. Effective feature extraction should encapsulate taxonomic information that can be feasibly acquired from the provided images [53]. Traditional image recognition methods primarily depend on manual image analysis for feature extraction. Recently developed feature extractors can draw out basic low-level image features from digital images, with substantial progress in expressing color, texture, and shape features. These feature descriptors for images, constituted by feature vectors, summarize the attributes of an image, with assistance from data related to color, shape, and texture. Low-level image features act as powerful tools for characterizing images [54]. 

In the context of content-based image retrieval, low-level attributes including color, texture, and shape are extensively employed for the purpose of feature extraction. Color, as a global feature, mirrors the surface characteristics of objects within corresponding image regions. Notably, color stands out as one of the most intuitive and salient image characteristics, exhibiting resilience against fluctuations in noise, image dimensions, orientation, and resolution [54]. Metrics such as color moments find utility in distinguishing images based on their color attributes [55,56,57]. Geometric properties can be deduced through the computation of parameters such as area, convex area, eccentricity, major and minor axis lengths, perimeter, solidity, equivalent diameter (sqrt(4*area/pi)), extent, and orientation [58,59]. Texture features, on the other hand, constitute another indispensable fundamental attribute, enabling deeper insights into the macroscopic qualities and subtle structures of an image. They provide a richer source of image information compared to color and shape features in isolation.

The sibling species of *Tribolium*, with their more vibrant colors, more regular shape edges, and smaller disparity between males and females, are easier to extract from wheat-based backgrounds compared to the *Sitophilus* family. *Sitophilus* species, being hidden insects, often burrow or hide within wheat grains, making it challenging to capture complete appearance information compared to secondary insects. However, insect posture and shooting angle can influence these finely extracted features. For instance, light conditions easily affect color and have poor noise immunity. Moreover, the lack of specific shooting parameters, such as the distance to the subject being photographed, frequently poses a difficulty in accurately determining the size of insects from images. Field images, due to their variable posture, diverse backgrounds, and different photographic conditions, pose a substantial challenge in processing. Taxonomy features are often concealed by posture or viewpoint, making detections based on the laboratory dataset consistently superior to those based on live datasets.

Among the natural features observed in the images, the most distinctive characteristic is the shape of the pronotum. The pronotum of the red flour beetle is rectangular, slightly rounded on the sides, with bluntly rounded anterior angles [60]. On the other hand, the pronotum of the confused flour beetle is also rectangular but with raised sides and sharply pointed anterior angles [27]. These differences in the shape of the pronotum serve as key distinguishing features between the two species.

However, texture represents merely a facet of an object’s surface and does not encompass the entirety of its physical attributes. Relying solely on texture features might not offer a comprehensive understanding of image content. Moreover, these features can demonstrate notable variances when there is a shift in image resolution [54]. Describing and obtaining the specific feature mentioned using computer language is challenging. Deep learning approaches differ from traditional object detection and recognition methods because they do not rely on prior knowledge to extract manually designed features. In contrast, they employ Multilayer Convolutional Neural Networks to acquire hierarchical representations of basic features (such as edges and corners) and subsequently integrate them, either linearly or through non-linear operations, to create more abstract high-level features. These features are automatically extracted and selected based on the content of the images. Deep learning neural networks can distinguish objects in various backgrounds, resolutions, and lighting situations after proper training without needing separate training for each potential characteristic. The pests occupy a very small proportion of the image. In an image with a 640 × 480 pixels resolution, the *Tribolium* label range falls between 16^2^ and 48^2^, indicating the limited visual information of such small scales. As convolutional layers increase, discriminative feature information and positional information gradually diminish, and the background becomes the main source of interference. Therefore, deeper convolutional layers are not necessarily better. This is particularly crucial for similar species like the red flour beetle and the confused flour beetle. However, the recognition process requires a combination of low-level features that differentiate the species and deep-level features that capture species semantics.

We have re-designed the size of the anchor boxes and the method of target localization. If the anchor boxes are too large, they provide limited contextual information about the targets. However, if they are too small, they may not contain enough discriminative information for recognition. The body length of the confused flour beetle adults ranges from 3 to 4.5 mm [27], while that of the red flour beetle adults falls between 2.3 and 4.4 mm [60]. The aspect ratio of their bodies is around 0.3, and the aspect ratio of the labels in the images ranges from 0.3 to 2.2. Compared to the standard setting of anchor boxes, our targets are very small and have a certain degree of overlap. In model training, we reset the scale of the anchor boxes to 8^2^, 16^2^, 24^2^, 32^2^, and 48^2^, increasing the number of anchor boxes that match the grain pests. This allows the model to detect more small targets. Although the quantity of intersecting bounding boxes may rise, the subsequent utilization of Soft Non-Maximum Suppression (Soft-NMS) efficiently decreases the prevalence of intersecting boxes, thus lowering the potential for missing targets. This is particularly valuable in scenarios featuring adhesive tar-like substances.

### 4.2. Data

There are various limitations to using deep learning approaches for visible light imaging-based detection and identification of stored-grain pests in practical warehouse management, including dataset size, sample diversity, and image quality. It requires the collection of clear and diverse sample images for model training. Typically, researchers use single-background images captured in the laboratory, such as specimens photographed or temporarily frozen pests in a pseudo-death state. While controlling pests in the laboratory setting allows for obtaining the desired image data, such data contains significant artificial interference factors and does not reflect the overall activity characteristics of the test sample population in the actual warehouse. Additionally, it does not meet the application requirements for pest recognition in real-world warehouse operations. It differs from face or vehicle recognition, which involves identifying specific feature differences within the same species. It recognizes stored-grain pests, notably for distinguishing closely related species and distinct geographical strains. We need to obtain statistical characteristics of the recognized sample population while minimizing the individual variation features exhibited by each sample. In this study, we utilized surveillance devices in the warehouse to capture images of grain pests in their natural state. Compared to laboratory-simulated image capture, the images obtained through interruption-free tracking in the warehouse environment contain richer variations in posture, which are beneficial for training the statistical characteristics of the objects to be recognized. After adaptive learning through Multilayer Convolutional Neural Networks, these features form comprehensive information about the natural postures of the objects without any artificial interference factors.

In practical applications, images from different surveillance devices may have variations in resolution. Through extensive experimental research, we have found that the data used for training is not just about quantity but rather about incorporating effective data that carry different morphological and resolution characteristics of a particular species. In our experiments using real warehouse data, the number of samples for red flour beetles is 15,138, which is significantly larger than the 9894 samples for confused flour beetles. However, the Average Precision (AP) value for confused flour beetles is 1.9% higher than red flour beetles in the system’s detection and recognition results. This indicates that data quantity is merely the foundation for model training, while effective data is the guarantee for improving detection and recognition performance. This also addresses a misconception about using images for pest recognition, i.e., how to extract the same features from images captured under different control conditions. For example, the impact of different lighting conditions on the texture of pests’ backs. Here, we have managed the image capture procedure to guarantee that images are taken in comparable or identical circumstances, extracting visually consistent distinguishing characteristics for a specific species. Training deep neural networks on images is similar to the human process of recognizing objects, which requires many images describing and labeling all object attribute information so that the system can detect and recognize a wide range of species and environments. The more variations of a particular object in different scenes are observed, the more distinctive its appearance becomes when encountering a new appearance. Collecting images using on-site surveillance devices holds promise for establishing a standardized dataset for detecting and recognizing stored-grain pests in real warehouse settings.

### 4.3. Advantages of Deep-Learning-Based Target Detection for Storage Grain Pests in Complex Backgrounds

Traditional image classification techniques based on visible light imaging struggle to identify small stored-grain pests, particularly those with similar colors and shapes, such as closely related species. Typically, microscopic magnification is required to capture images of stored-grain pests, followed by edge extraction algorithms to obtain recognizable outlines or calculating gray-level co-occurrence matrices to capture different texture features [61]. However, even with these methods, recognition accuracy can be compromised due to intra-species differences or the small size of individuals in complex backgrounds. 

Figure 11a compares three sets of edge feature extraction for *Tribolium* species. The first set represents data captured using a smartphone, the second set represents data captured through microscopic magnification, both with filter paper backgrounds, and the third set represents data collected using surveillance equipment with grain as the background. The samples can be separated from the background through edge feature extraction in the first and second sets. Different species can be distinguished based on the numerical values of these edge features. In the second set of images captured through microscopic magnification, the confused flour beetles display more prominent edges on their elytra, indicating that when the samples are magnified to a certain extent, more information about the detection target is captured. It becomes possible to identify differences between closely related species in the images. However, computer calculations cannot quantify or automatically extract these irregular features. In the third set, where the background consists of grains, separating and identifying stored-grain pests through edge extraction is not feasible.

Therefore, recognizing stored-grain pests based on image analysis often relies on hyperspectral or multispectral imaging techniques [7,62,63], which can identify spectral characteristics not visible to the naked eye or ordinary cameras. However, even with these specialized spectral imaging technologies, handling images with a background of real storage facility grains containing multiple grain pest targets is challenging.

Since the network learns to automatically extract and choose relevant properties through numerous annotated target photos, object detection based on deep learning eliminates the need to examine image features manually. A trained deep learning model can effectively identify objects in different backgrounds, resolutions, and lighting conditions. Figure 11b illustrates the automatic feature extraction process used by the network on annotated target features with a background of grains. Experimental results have demonstrated that deep-learning-based object detection methods have advantages in extracting features and classifying stored-grain pests in complex backgrounds, relying on a substantial amount of annotated data. 

### 4.4. Advantages in Food Safety and Grain Quality Preservation

Pests in grain storage facilities not only consume the grains but also release moisture. This, in turn, can lead to mold development, thereby jeopardizing the quality of the stored grain. Detecting such infestations early on is of paramount importance. Intriguingly, insects can reach a density of 1000 to 2000 per ton before becoming visible, causing a significant decline in the food products’ commercial value [64,65]. To put this in perspective, almond kernels infested with pests can witness a dramatic price drop from $2 per pound to just $0.2. Similarly, if milled rice shows a mere 2.5% reduction in quality because of these pests, it can translate to notable financial setbacks, approximately $5.5 for every ton of paddy rice, as supported by previous studies [66]. The process of image recognition and interpretation plays a pivotal role by providing valuable insights derived from image analysis. These insights can be effectively applied in controlling various processes or machinery [67]. Notably, this technology has demonstrated its efficacy in monitoring and evaluating diverse aspects of bulk grains. These aspects encompass the identification of grain varieties, assessment of their overall quality and integrity, detection of foreign objects, identification of mold presence, and evaluation of insect infestation levels [68]. Moreover, machine vision technology is notably well-suited for the detection of both entire insects and live insects within stored-grain samples. For instance, in a study conducted by Zayas and Flinn (1998) [69], the utilization of multispectral analysis and pattern recognition techniques led to the successful detection of R. dominica in bulk wheat samples. Impressively, this approach achieved recognition rates exceeding 90% for adult lesser grain borers and certain foreign objects.

In essence, our exploration in this field is not just confined to entomology. We are actively contributing to heightening food safety standards and maintaining quality in the grain storage and processing sectors. By incorporating cutting-edge image recognition methodologies into grain storage procedures, our goal is to minimize the adverse effects of insect pests on food availability and economic outcomes.

## 5. Conclusions

This study delves deeper into the automatic identification of closely related insect species using visible light imaging technology and significantly improves the previous models. Along with the neural network architecture, the rational data structure enhances the efficiency of detection and identification. The new model shows approximately 0.133 s/img and 0.055 s/img improvement in prediction speed for *Sitophilus* and *Tribolium*, respectively. It achieves 2.32% and 2.53% higher recognition rates for *Sitophilus* and *Tribolium* than the previous model, respectively, even in distinguishing between different geographical strains. Furthermore, by integrating advanced image recognition techniques into the grain storage process, we aim to mitigate the impact of insect pests on food security and economic losses. Our work underscores the critical importance of early detection and intelligent pest management in preserving the integrity of food products within the grain industry.

## Figures and Tables

**Figure 1 foods-12-03653-f001:**
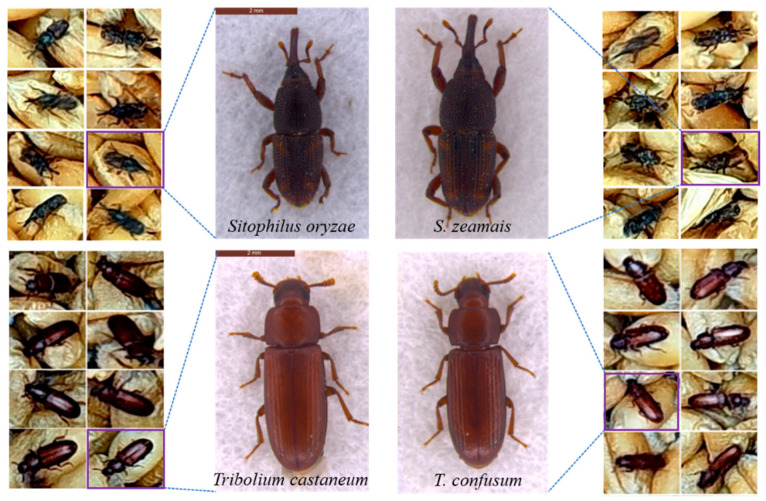
Visual assessment of grain damage by *Tribolium* and *Sitophilus* sibling species.

**Figure 2 foods-12-03653-f002:**
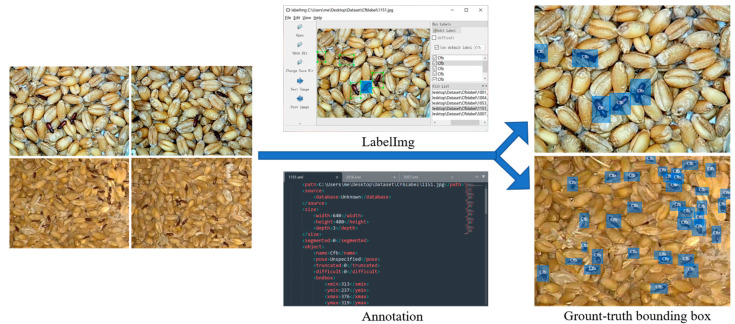
Labelling target object.

**Figure 3 foods-12-03653-f003:**
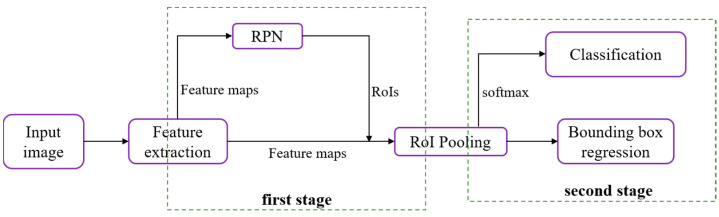
The structural design of the two-stage approach, featuring the Region Proposal Network (RPN) and Region of Interest (RoI).

**Figure 4 foods-12-03653-f004:**
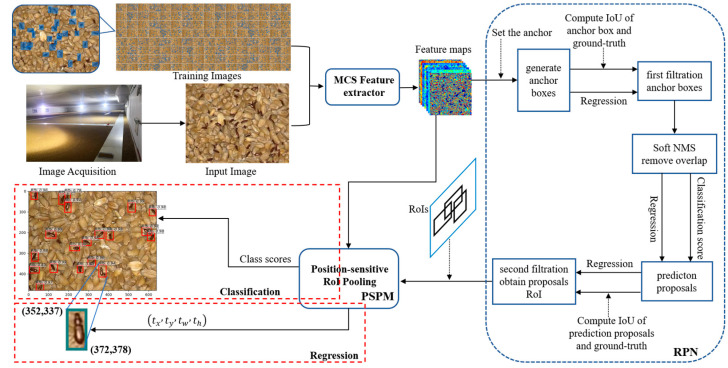
The design structure of the proposed MCSNet+ framework. MCS: Multilayer Convolutional Structure, RPN: Region Proposal Network, PSPM: Position-Sensitive Prediction Module, RoI: Region of Interest, IoU: Intersection over Union, Soft NMS: Soft Non-Maximum Suppression.

**Figure 5 foods-12-03653-f005:**
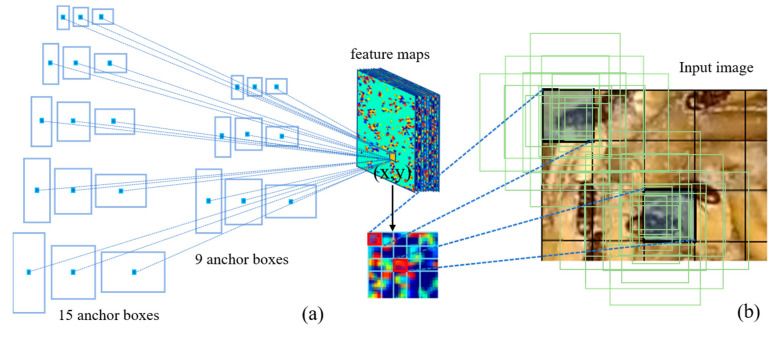
Schematic of anchor box diagram. (**a**) Illustrates the various sizes and aspect ratios of anchor boxes corresponding to a sample point (x,y)-coordinate on the feature map; (**b**) Showcases how different sample point (x,y)-coordinates on the feature map are mapped back to the anchor boxes in the original image.

**Figure 6 foods-12-03653-f006:**
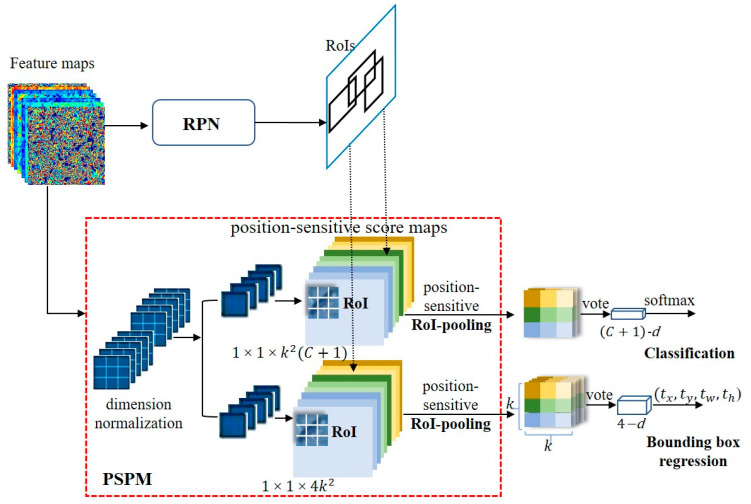
Structure of the Position-Sensitive Prediction Module (PSPM). ‘d’ denotes the dimension.

**Figure 7 foods-12-03653-f007:**
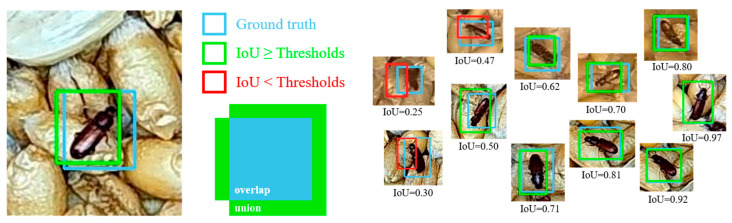
Overlap area and union area of ground truth and prediction.

**Figure 8 foods-12-03653-f008:**
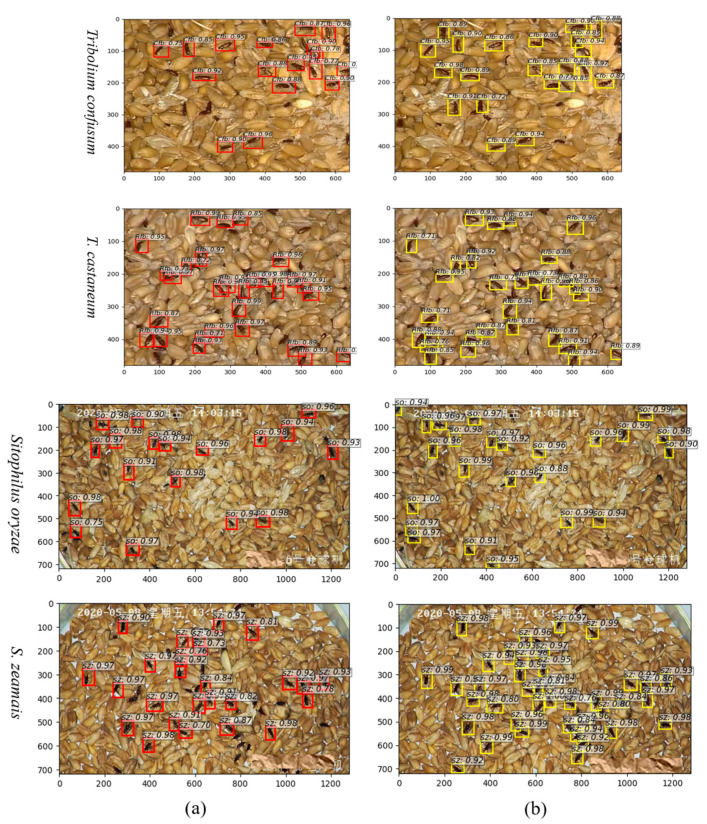
Visualization of the detection and recognition outputs in different anchor scales: (**a**) Represents 8^2^, 16^2^, and 32^2^; (**b**) Represents 8^2^, 16^2^, 24^2^, 32^2^, and 48^2^. SO: *Sitophilus oryzae*, SZ: *S. zeamais*, Rfb: *Tribolium castaneum*, Cfb: *T. confusum*.

**Figure 9 foods-12-03653-f009:**
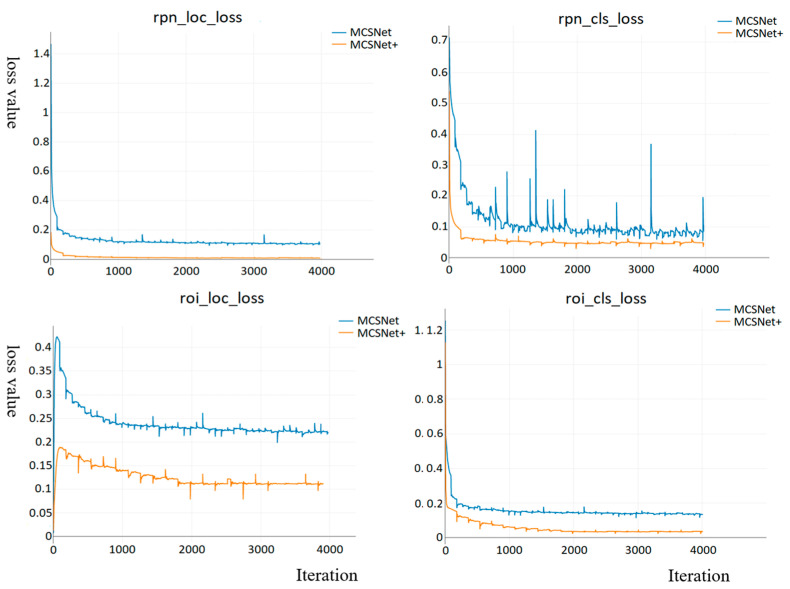
The classification loss and regression loss curves of MCSNet and MCSNet+ on the training set for the Region Proposal Network (RPN) and Region of Interest (RoI). rpn_cls_loss: Region Proposal Network classification loss, rpn_loc_loss: Region Proposal Network location loss, roi_cls_loss: Region of Interest classification loss, roi_loc_loss: Region of Interest location loss.

**Figure 10 foods-12-03653-f010:**
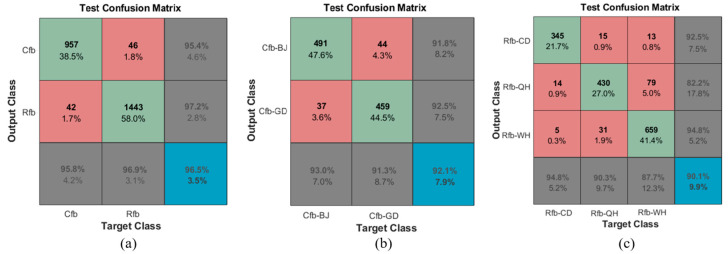
Confusion matrixes in MCSnet+ with the *Tribolium* filed dataset: (**a**) Two sibling species of *Tribolium*; (**b**) Two geographical strains of *T. confusum*, namely Cfb-BJ and Cfb-GD; (**c**) Three geographical strains of *T. castaneum*, specifically Rfb-CD, Rfb-QH, and Rfb-WH. The green cells indicate correct predictions, while the red cells represent incorrectly predicted samples. The blue cell in the bottom right displays the total percentage of accurately predicted samples alongside the total incorrectly predicted samples. The numbers at the top of the cell represent the number of predicted samples, while the accompanying percentage illustrates their proportion within the total number of positive test samples.

**Figure 11 foods-12-03653-f011:**
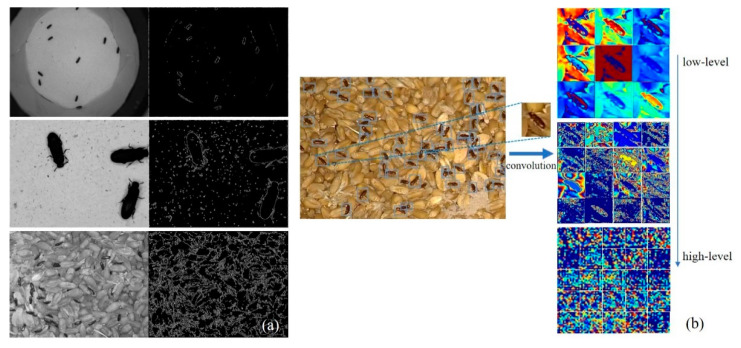
Comparison of traditional machine learning feature extraction and deep-learning-based feature extraction. (**a**) Compares three sets of edge feature extraction; (**b**) Illustrates the automatic feature extraction.

**Table 1 foods-12-03653-t001:** Comparative analysis of four network architectures—VGG16, ResNet50, ResNet101, and MCS—for automated identification of *Tribolium* species, across four distinct training and testing configurations.

Feature Extractor	Pretrain/ Test Data	Sample Size			AveP(Cfb) Mean ± Std (%)	AveP(Rfb) Mean ± Std (%)	mAP Mean ± Std (%)	Predict Speed Mean ± Std (s/img)
		Training (72%)	Validation (18%)	Test (10%)				
VGG16	ImageNet/ Field *Tribolium* W.	17,975 ± 112	4516 ± 86	2499 ± 89	87.17 ± 1.16	87.38 ± 2.33	87.27±1.36	0.327±0.016
ResNet50		17,975 ± 112	4516 ± 86	2499 ± 89	88.64 ± 1.95	84.71 ± 3.75	86.68 ± 1.92	0.385 ± 0.027
ResNet101		17,975 ± 112	4516 ± 86	2499 ± 89	84.88 ± 4.19	85.50 ± 2.73	85.17 ± 3.08	0.402 ± 0.021
MCS		17,975 ± 112	4516 ± 86	2499 ± 89	88.79 ± 2.43	87.28 ± 1.65	88.03 ± 1.52	0.319 ± 0.015
VGG16	ImageNet/Lab+Field *Tribolium* W.	26,591 ± 220	6688 ± 66	3657 ± 265	92.62 ± 0.93	91.03 ± 0.98	91.83 ± 0.84	0.274 ± 0.011
ResNet50		26,591 ± 220	6688 ± 66	3657 ± 265	92.91 ± 0.90	89.64 ± 1.22	91.28 ± 0.83	0.369 ± 0.014
ResNet101		26,591 ± 220	6688 ± 66	3657 ± 265	91.34 ± 1.09	89.60 ± 1.31	90.47 ± 0.99	0.358 ± 0.015
MCS		26,591 ± 220	6688 ± 66	3657 ± 265	92.99 ± 1.01	91.27 ± 0.76	92.14 ± 0.67	0.268 ± 0.006
MCS	Lab SOSZ W./ Field *Tribolium* W.	17,975 ± 112	4516 ± 86	2499 ± 89	92.78 ± 0.89	90.69 ± 0.77	91.74 ± 0.82	0.281 ± 0.008
	Lab SOSZ W./Lab+Field *Tribolium* W.	26,591 ± 220	6688 ± 66	3657 ± 265	93.34 ± 1.12	91.85 ± 0. 81	93.90 ± 0.87	0.259 ± 0.006

Abbreviations: AveP, Average Precision; mAP, mean Average Precision; std, standard deviation.

**Table 2 foods-12-03653-t002:** Evaluation of various anchor scales with a fixed aspect ratio (1:1, 1:2, and 2:1) on the MCS architecture.

Anchor Scales	Data	RoI Generation Speed Mean ± Std (s/img)	Predict Speed Mean ± Std (s/img)	mAP Mean ± Std (%)
MCS_-8,16,32_	Field *Tribolium* W.	0.133 ± 0.001 b	0.285 ± 0.004 c	88.40 ± 0.83 c
MCS_-4,8,16,32_		0.150 ± 0.016 b	0.335 ± 0.031 bc	88.92 ± 0.99 bc
MCS_-8,16,24,32,48_		0.148 ± 0.020 b	0.338 ± 0.027 bc	89.27 ± 0.79 b
MCS_-8,16,24,32,48,56_		0.151 ± 0.012 b	0.360 ± 0.055 b	88.50 ± 0.70 bc
MCS_-4,8,16,24,32,48_		0.225 ± 0.077 a	0.460 ± 0.126 a	90.40 ± 0.43 a

Note: lowercase shows significant difference at P_0.05_.

**Table 3 foods-12-03653-t003:** The detection and recognition of labeled insect samples in the test set with the MCSNet+ model.

Model	Data	Category	Test Sample Size	Precision Mean ± Std (%)	Recall Mean ± Ste (%)	AveP Mean ± Std (%)	mAP Mean ± Std (%)	Predict Speed Mean ± Std (s/img)
MCSNet+	Lab.	Cfb	709 ± 48	92.10 ± 1.44	98.31 ± 0.43	96.83 ± 0.42	96.83 ± 0.43 a	0.193 ± 0.010 a
		Rfb	481 ± 26	94.08 ± 2.36	97.37 ± 0.85	96.82 ± 0.51		
		SO	244 ± 7	93.46 ± 1.66	97.27 ± 2.25	93.89 ± 3.63	94.36 ± 3.28 a	0.204 ± 0.003 a
		SZ	182 ± 28	94.55 ± 2.40	97.07 ± 1.53	94.82 ± 3.29		
	Field	Cfb	990 ± 182	95.04 ± 2.42	95.76 ± 1.97	95.22 ± 1.47	94.27 ± 1.02 b	0.226 ± 0.004 b
		Rfb	1531 ± 168	94.37 ± 2.36	96.88 ± 2.22	93.31 ± 1.59		
		SO	1079 ± 16	93.25 ± 2.36	94.16 ± 4.19	91.93 ± 1.23	92.67 ± 1.74 a	0.240 ± 0.004 b
		SZ	1367 ± 10	95.31 ± 0.94	97.50 ± 0.97	93.42 ± 2.69		
	Lab+Field	Cfb	1698 ± 174	95.22 ± 1.29	97.10 ± 1.79	95.49 ± 0.86	95.29 ± 0.55 c	0.205 ± 0.005 c
		Rfb	2012 ± 163	94.82 ± 1.53	95.81 ± 1.49	95.09 ± 0.68		
		SO	1323 ± 17	90.59 ± 2.59	93.75 ± 1.16	94.38 ± 0.74	94.94 ± 0.70 a	0.227 ± 0.005 c
		SZ	1550 ± 13	92.51 ± 1.27	94.88 ± 0.83	95.50 ± 0.76		

Note: lowercase shows significant difference at P_0.05_.

**Table 4 foods-12-03653-t004:** Detection of *Tribolium* geographic strains based on MCSNet+ with field dataset.

Geographical Strains	Test Samples Size	Precision Mean ± Std (%)	Recall Mean ± Ste (%)	AveP Mean ± Std (%)	Mean AveP Mean ± Std (%)
Cfb-BJ	509 ± 80	88.4 ± 2.02	92.97 ± 0.97	85.04 ± 2.51	84.71 ± 1.33
Cfb-GD	480 ± 114	87.83 ± 1.04	91.47 ± 1.71	84.38 ± 1.63	
Rfb-CD	354 ± 94	91.53 ± 0.89	94.84 ± 1.37	89.74 ± 1.48	86.28 ± 1.20
Rfb-QH	452 ± 125	90.97 ± 1.42	90.30 ± 2.40	86.46 ± 2.15	
Rfb-WH	704 ± 171	89.85 ± 1.25	87.76 ± 1.34	82.64 ± 1.27	

## Data Availability

The data presented in this study are available on request.

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
