# Peer review of "MCSNet+: Enhanced Convolutional Neural Network for Detection and Classification of Tribolium and Sitophilus Sibling Species in Actual Wheat Storage Environments"

_foods, 2023, doi:10.3390/foods12193653_

Round 1

Reviewer 1 Report

Comments and Suggestions for Authors

This study presents the development of an improved automated model for the detection and classification of Tribolium and Sitophilus, through convolutional neural networks, as a tool that provides valuable information for management decision-making in grain storage facilities. The work is a significant contribution, which presents advances in an area of ​​great potential for pest management in the future. It is correctly written and structured in its different sections. There are a few form fixes that are included in the attached file. I consider that the work needs minor corrections.

Author Response

Dear Prof.,

Thank you for your expedient response and suggestions. We have addressed all comments from you. We found the comments helpful and hope that we have incorporated the suggested improvements to your satisfaction? The list of addressed review comments is attached as below.

Yours sincerely

Yanyu Li

Reviewer 2 Report

Comments and Suggestions for Authors

General comments

1.      The MS is important since it targets the problem of correct identification of siblings -i.e. mostly congeneric species. However, it is not well written and planned. As a result, an independent reader would be greatly confused by the structure and jargon used.

2.      The pest status and the pesticide resistance of the two beetle species are well known. However, the authors do not adequately describe how the taxonomic distinction of sibs helps in the problem of their confrontation. Suddenly in line 115 of the MS, the reader reads, "Previous studies indicated that MCSNet correctly identified two Sitophilus sibling 115 species on stored wheat”. The problem of correct identification is essential for entomologists and there is no urgent need for other applications where the taxonomic separation would offer additional benefits.

3.      Nevertheless, the problem of CNN-based identification of the two sibling species would offer important services to business and political authorities with no specialized entomological personnel. Of course, this is the main task of Artificial Neural Networks. They offer expertise services in cases of lacking specialized personnel.

4.      This MS suffers from the disadvantage of all AI methods so far. The lack of a reduced set of features that differentiate sibs at a specific age. Such a future is known to be the structure of the antennal apex, which is the tripartite club in T. castaneum and the quadripartite gradual club in T. confusum.

5.      To facilitate readers I suggest the authors provide an account of the mathematical procedures and involved programs in a separate supplementary note/figure.

Specific comments

6.      Line 129: It is unclear what the authors mean by “strain”. Is it a geographical race, an ecotype, a variety, or a population? Usually, the term ‘strain’ is in use for fungi or bacteria. If an identification work is described then the authors should be cautious about using the terminology of another discipline.

7.      L.154: Please correct the period sign.

8.      L.157-163: A sample of the image database would offer additional insight to readers. Especially when the images include a variable number of Tribolium sp. Individuals.

9.      L.161-163: An example figure would greatly facilitate reading the MS. In addition, the availability of the program should be given.

10.   L.160: With 200 images per geographic “strain” (=population) the problem of siblings identification should be additionally difficult since the inter-population variation poses additional problems. The authors seem not to get annoyed by that source of diversity.

11.   L.206: I applaud the authors to gather and write all abbreviations and acronyms in a separate session of the MS in order to facilitate readers.

12.   L. 205: No reference is cited for 37 (bibliographic entry). In effect, ‘Faster R-CNN’ seems to be the invention of the authors.

13.   Table 4: Should be rewritten taking into account the spacing of numbers.

14.   L.655-657: The auxiliary use of this method would be justified if the differentiating features among siblings were predicted. Otherwise, it is useless. Moreover, the au

15.   thors of reference 43 revealed differences in the biology between geographical races of T. confusum only. Not morphological ones in both sibs. In addition, the genetic basis of the revealed different phenologies between geographical races of T. confusum is only speculated not proved.

16.   L.626-632: Please move to M&M session.

17.   (deleted)

18.   Table 3: An explanation is needed for the numbers MCS-8,16,24,32,48. In addition, the bold face letters in the third row need an explanation in the legend of the table.

Author Response

(The authors gave the same response as above.)

Reviewer 3 Report

Comments and Suggestions for Authors

Title:

ok.

Abstract:

·         Technical Details on Models: While the abstract mentions the utilization of convolutional neural networks and specific enhancements such as Soft Non-Maximum Suppression and Position-sensitive prediction, a brief description of these techniques and how they contribute to the improved performance would enhance the abstract's technical clarity.

·         Conclusion Emphasis: In the final sentences, stress the implications of the successful achievement in terms of its contribution to establishing an online pest management system. This connection would reinforce the practical value of the research.

Keywords:

·         Please check the first keyword spellings.

Introduction

Species Description (lines 51-81): Briefly describe T. castaneum and T. confusum (lines 51-81), including their lifecycle and behavior.

Technical Clarity (lines 96-97): Explain technical terms like "Enhanced Graphics Module (Position sensitive prediction model)" (lines 96-97) for reader comprehension.

Research Gap and Objective (lines 118-125): State the innovative aspect of your approach and its specific objectives (lines 118-125).

Scope and Characters (lines 122-123): Specify traits to be evaluated for association with sibling species populations (lines 122-123).

Practical Benefits (lines 108-109): Highlight how outcomes can aid entomologists, businesses, and regulators (lines 108-109).

Figure Context (lines 111-114): Briefly explain figures' content when referenced in the text (lines 111-114).

Expansion on MCSNet (lines 115-117): Elaborate on the link between the current study and prior work involving MCSNet (lines 115-117).

Materials and Methods

Insects (lines 128-135): Defend rearing procedure with reference.

Data Collection (lines 138-156): Explain the two parts of the image data used: laboratory and field acquisitions. Provide context for the laboratory setup and how images were captured in the field.

Data Augmentation (lines 184-194): Describe the purpose of data augmentation to enhance the diversity and robustness of the dataset. Explain the transformations applied, such as rotations, flips, and changes in scale, along with their benefits.

Results:

Clear Abbreviations (Lines 516-517): Clarify abbreviations for geographical strains in the dataset names (Cfb-BJ, Cfb-GD, etc.) to avoid confusion for readers.

Consistent Units (Lines 525-526, 529-530, etc.): Maintain consistent units of measurement throughout the section and specify decimal places (e.g., mean±std(s/img)) for reported values.

Loss Function Variability (Lines 580-581): Explain the significance of fluctuating loss functions and their impact on model convergence and performance.

Geographical Strain Variations (Lines 608-609): Provide practical context for performance variations among geographical strains and their implications for pest management.

Discussion:

Lines 659-665: Clarify the importance of effective feature extraction for accurate identification results and highlight recent advancements in feature extractors.

Lines 682-693: Discuss challenges related to insect characteristics, posture, and shooting angle on feature extraction, especially for difficult species like Sitophilus.

Lines 694-699: Highlight the significance of the pronotum shape as a distinguishing feature between species.

Lines 736-741: Discuss the limitations of using laboratory images for training due to artificial interference factors and the benefits of on-site surveillance.

Lines 742-753: Emphasize the importance of effective data in influencing detection and recognition performance.

Lines 782-794: Briefly discuss how deep learning overcomes the challenges of traditional image classification for stored-grain pests.

Lines 808-815: Highlight the advantages of deep learning-based object detection, particularly in handling complex backgrounds.

Lines 821-830: Summarize key findings about improved identification accuracy and practical applications of the developed model.

General comments

 After carefully reviewing the manuscript, I must commend the author for their skillful writing and overall presentation. However, I have identified several areas where the manuscript could be improved. These suggestions are intended to help the author further enhance the manuscript's readability, structure, and impact.

Comments on the Quality of English Language

Careful proofreading is required. 

Author Response

(The authors gave the same response as above.)
